# Pharmacological Topical Therapy for Intra-Oral Post Traumatic Trigeminal Neuropathic Pain: A Comprehensive Review

**DOI:** 10.3390/ph17020264

**Published:** 2024-02-19

**Authors:** Yair Sharav, Shimrit Heiliczer, Rafael Benoliel, Yaron Haviv

**Affiliations:** 1Department of Oral Medicine, Sedation and Imaging, Hadassah Medical Center, Faculty of Dental Medicine, Hebrew University of Jerusalem, Jerusalem 91120, Israel; sharavy@mail.huji.ac.il (Y.S.); hsaroni@yahoo.com (S.H.); 2Oral Medicine Unit, Oral and Maxillofacial Surgery Department, Tel Aviv Sourasky Medical Center, Tel Aviv 64239, Israel; 3Rutgers School of Dentistry, Newark, NJ 07103, USA; benolielr@gmail.com

**Keywords:** neuropathic pain, intra-oral pain, topical pain treatment, post-traumatic neuropathy, neurosensory stent

## Abstract

**Background**: The efficacy of topical treatments in alleviating neuropathic pain is well-established. However, there is a paucity of research on topical interventions designed specifically for intra-oral application, where the tissue composition differs from that of exposed skin. **Methods**: This comprehensive review endeavors to assess the extant evidence regarding the efficacy of topical treatments in addressing neuropathic pain within the oral cavity. Utilizing combinations of search terms, we conducted a thorough search across standard electronic bibliographic databases—MEDLINE (via PubMed), Embase, Google Scholar, and Up to Date. The variables under scrutiny encompassed topical treatment, local intervention, chronic oral and orofacial pain, and neuropathic pain. All pertinent studies published in the English language between 1992 and 2022 were included in our analysis. **Results**: Fourteen relevant manuscripts were identified, primarily consisting of expert opinions and case reports. The comprehensive review suggests that topical treatments, especially when applied under a stent, could be effective in mitigating neuropathic pain in the oral area. However, it is crucial to conduct further studies to confirm these preliminary results. The limitations of the reviewed studies, mainly the reliance on expert opinions, small sample sizes, inconsistent study designs, and a lack of long-term follow-up data, highlight the need for more rigorous research. **Conclusions**: Although initial findings indicate topical treatments may be effective for oral neuropathic pain, the limitations of current studies call for more thorough research. Further comprehensive studies are essential to validate the efficacy of these treatments, standardize procedures, and determine long-term results. This will provide clearer guidance for treating chronic neuropathic pain in the oral cavity.

## 1. Introduction

Topical drug application delivers a high concentration of the drug locally at the pain site with minimal systemic effects, reducing drug interactions, which is beneficial for patients on multiple medications or those avoiding specific side effects, such as NSAIDs causing gastric ulcers [1,2]. Despite its advantages, there is limited research on topical treatments for the unique tissue environment of the intra-oral area. This review seeks to shed light on topical therapies for chronic oral neuropathic pain.

Topical agents must be able to cross the epithelium or mucosa to induce an effect. The effectiveness of this barrier is dependent on its integrity and age-related and disease-related changes. Depending on thickness and keratinization, the oral mucosa is up to 10 times more permeable than the skin, allowing rapid drug penetration [1]. The oral mucosa is, therefore, very suitable for topical drug application. However, the most resistant and the least permeable mucosa is known as “*masticatory*”; it receives daily friction, resulting in the keratinization of its epithelium. It covers the hard palate and gingiva, and passing through it may create a challenge. Additionally, the effects of saliva will limit the contact time between the drug and mucosa and may significantly limit efficacy. Therefore, the use of topical agents intraorally requires either prolonged isolation of the area or the construction of an intra oral appliance (stent) that will allow optimal concentrations for an adequate period [1]. There are various commercially available topical analgesics, mostly single drugs, but some combinations are available. Many practitioners use single drugs in combination or have them specifically compounded into gels or creams by specialized pharmacies [2]. The following review aims to address the treatment of intraoral neuropathic pain with an intraoral stent, commonly referred to as a neurosensory stent (NS). Orofacial neuropathic pain types will be discussed, followed by the construction of an NS and the medications used for the local topical treatment of post-traumatic trigeminal neuropathic pain in the oral cavity. We conducted a thorough search across standard electronic bibliographic databases—MEDLINE (via PubMed), Embase, Google Scholar, and Up to Date. The variables under scrutiny encompassed topical treatment, local intervention, chronic oral and orofacial pain, and neuropathic pain. All pertinent studies published in the English language between 1992 and 2022 were included in our analysis.

## 2. Orofacial Neuropathic Pain Types

Neuropathic pain occurs as a result of damage or injury to the somatosensory system and can cause significant suffering, resulting in severe limitations to daily activities and, ultimately, diminishing the overall quality of life for patients [3,4,5]. In the general population, the incidence of neuropathic pain is estimated to be approximately 7–8%, accounting for about 20–25% of individuals who suffer from persistent pain [5]. In the orofacial region, neuropathic pain can have multiple origins. It may stem from central nervous system disorders, such as trigeminal neuralgia [6]. Alternatively, damage to the peripheral nervous system, often associated with dental procedures or other forms of trauma, can lead to conditions like post-traumatic trigeminal neuropathic pain [7]. Viral infections are another potential cause, as with post-herpetic neuralgia [8]; another condition is persistent idiopathic facial pain (PIFP), chronic pain of a neuropathic nature without any known cause [9,10].

Chronic pain, especially when it takes the form of neuropathic pain that is both long-lasting and challenging to treat, has a profound impact on human health and well-being. This kind of pain frequently causes considerable emotional distress, manifesting as anxiety and depression, and it significantly hampers everyday activities and functioning [11]. The persistent and resistant nature of this pain highlights the critical necessity for effective management strategies to alleviate its detrimental effects on individuals’ lives.

This review focuses on evaluating topical treatments for intra-oral neuropathic pain, particularly post-traumatic trigeminal neuropathic pain, excluding other forms of general, orofacial, or oral pain. The rationale for this review stems from the need to understand the effectiveness of such treatments in managing a specific subset of neuropathic pain, identifying gaps in current research, and highlighting the importance of addressing these gaps to improve patient outcomes.

### Post-Traumatic Trigeminal Neuropathic Pain and Its Management

The ICOP (International Classification of Orofacial Pain), 1st Edition, describes post-traumatic trigeminal neuropathic pain as “a unilateral or bilateral facial or oral pain following and caused by trauma to the trigeminal nerve(s), with other symptoms and/or clinical signs of trigeminal nerve dysfunction, and persisting or recurring for more than 3 months” [12]. The severity of nerve injuries may range from mild to severe. They include external trauma and iatrogenic injuries from dental treatments such as local anesthetic injections, root canal therapies, extractions, oral surgery, dental implants, orthognathic surgery, and other invasive procedures [13].

The pathophysiology of post-traumatic trigeminal neuropathic pain includes both peripheral and central mechanisms. Nerve injury, whether due to trauma or disease, triggers inflammatory processes, modifies the electrical activity of neurons, and facilitates cross-talk between neurons, leading to a phenomenon known as peripheral sensitization [14]. The release of inflammatory mediators leads to the modified expression and distribution of sodium channels at the site of injured nociceptors and their associated ganglia. Furthermore, calcium channels and adrenergic receptors may be expressed in these axons, resulting in a reduced nociceptor depolarization threshold and the occurrence of ectopic discharges [15]. These characteristics of nerve injury could contribute to the spontaneous pain observed in patients with post-traumatic trigeminal neuropathic pain [14]. The afferent barrage from peripheral nociceptors induces central sensitization, a phenomenon involving structural changes in the connectivity of second- and third-order neurons in the central nervous system [16].

Notwithstanding the extensive prevalence of dental procedures capable of inducing trigeminal nerve injury, it is noteworthy that a majority of patients do not exhibit postoperative pain. Nevertheless, a subset, approximately 3–5%, develop chronic pain following injury to trigeminal nerve branches [17,18], attributable to a range of etiologies from significant traumas such as zygomatic fractures to relatively minor dental procedures like root canal therapy or dental implant placement. Post-traumatic trigeminal neuropathic pain is typically characterized by its unilateral presentation, restricted to the dermatome correlating with the site of nerve injury. The pain profile of this neuropathy is often delineated as burning, electrical, or stabbing in nature, frequently accompanied by either positive or negative sensory alterations [7,19]. It often leads to substantial patient distress [7]. Its therapeutic efficacy is commonly gauged by a reduction in pain intensity and/or frequency, with a benchmark of 50% or greater diminution being indicative of successful intervention [20,21]. In certain neuropathic pain scenarios, even a 30% alleviation of symptoms is considered clinically meaningful [20].

The treatment of pain involves abortive treatment during pain or prophylactic treatment regardless of pain. Neuropathic pain, which is usually chronic, may be better suited for long-term prophylactic treatment [22]. Systemic drug treatments are commonly used for managing neuropathic pain. The systemic drugs commonly used include antiepileptic medications such as pregabalin, antidepressants such as duloxetine, and tricyclic antidepressants such as amitriptyline [22]. Additional treatment options for pain control include medications such as lidocaine, anti-inflammatory drugs, α2 adrenergic receptor agonists, and NMDA receptor agonists [10]. However, the systemic administration of antiepileptic or antidepressant medications results in significant side effects [22].

The mechanism in which the systemic drugs work in managing neuropathic pain involves various pathways. Tricyclic antidepressants act as membrane stabilizers, preventing ectopic activity, and function as sodium channel and cholinergic receptor antagonists. Furthermore, they enhance the efficiency of the descending pain inhibitory system [23]. Anticonvulsants such as phenytoin and carbamazepine, as well as local anesthetics like lidocaine and mexiletine, primarily target sodium channels in neural tissue for their therapeutic effects. A newer generation of drugs, including topiramate, oxcarbazepine, and lamotrigine, has demonstrated activity in both sodium and calcium channel subtypes with fewer side effects. The analgesic properties of gabapentin and its related compound, pregabalin, seem to be linked to a strong affinity for a specific α2δ subunit of the voltage-gated calcium channel [16]. A summary of commonly used chronic pain medications for neuropathic pain, along with their modes of action, systemic adverse effects, and medical contraindications, is provided in Table 1.

## 3. Neuropathic Pain Topical Management

Topical therapy for pain, applied on bare skin, constitutes a prevalent modality in pain management. It encompasses the utilization of creams, gels, and patches laden with a multitude of analgesic agents. These topical therapies offer targeted relief for a range of conditions, such as arthritis, muscle strains, neuropathic pain, and more [25,26,27,28,29,30,31,32]. Currently, topical treatments are regarded as second or third-line options for patients with peripheral neuropathic pain. However, recently, it has been recommended that these agents be more broadly used as first-line treatments [28,33]. This has underscored the need to effectively communicate the advantages associated with topical agents.

The advantage of topical treatment is the straightforward and safe route of administration, rendering them an attractive option for patients with medical complications. These medications facilitate optimized drug concentration at the pain site, with minimal, if any, systemic absorption. For instance, in a comparison of systemic bioavailability between topical diclofenac sodium gel 1% and oral diclofenac sodium 50-mg tablets in healthy volunteers, it was found that systemic exposure with diclofenac sodium gel was up to 17-fold lower than with oral diclofenac. As a result, titration for achieving effective levels is not required. Moreover, topical formulations have high drug bioavailability at the application site since they avoid first-pass metabolism elimination within the gastrointestinal tract. Additionally, topical medications usually do not exhibit significant drug interactions and are associated with negligible side effects, barring occasional local allergies or rashes [1,33,34,35,36]. This facilitates long-term treatment, enhancing patient adherence and, consequently, the overall success of the treatment [37].

Local formulations may include agents commonly used orally or intravenously for neuropathic pain, such as lidocaine, antiepileptic drugs (e.g., gabapentin), antidepressants (e.g., amitriptyline), anti-inflammatory drugs, α2 adrenergic receptor agonists (e.g., clonidine), and NMDA receptor agonists (e.g., ketamine) [10,38]. The topical agents target local peripheral tissues to exert a therapeutic effect on peripheral receptors. However, our understanding of the precise mechanisms and the intricate cellular and tissue-level processes underlying the localized (topical) action of drugs that are traditionally used in systemic pain management is notably insufficient. While we recognize the efficacy of these drugs when administered systemically, targeting pain at its central neural pathways, our grasp of how these same agents exert their effects locally within the oral mucosa and surrounding tissues remains elusive for most of them.

Some topical treatments suggested for neuropathic pain on uninjured skin include lidocaine up to 5% three times a day (up to 250 mg of lidocaine base) and Capsaicin 0.025–0.075% paste up to four times a day for up to 8 weeks or an 8% patch—1 to 4 patches, for 30–60 min once every three months [23].

Importantly, this discussion excludes transdermal drug administration, which administers drugs systemically via the skin and is not considered topical [39].

### 3.1. Oral Neuropathic Pain Topical Management

In the orofacial region, akin to other body parts, topical medications are directly applied to the painful or affected site to initiate their pharmacological effects [40].

Various studies have demonstrated the efficacy of topical treatments in reducing orofacial pain [10,38]. A case report study demonstrated a decrease in orofacial neuropathic pain level and an enhancement in the quality of life with the sole use of topical lidocaine medicated plaster [41]. Another study demonstrated topical capsaicin efficacy in 12 patients with trigeminal neuralgia [42].

In some cases, optimal results may require a combination of multiple therapeutic agents. This is because active molecules applied in topical formulations exert multiple mechanisms of action, directly or indirectly modulating distinct molecular targets and pathways in the nociceptive system [25,38]. As a result, topical agents have the potential to reduce or even replace systemic medication, alleviating significant side effects. Moreover, its application can also be considered an “escape drug” or add-on method when pain exacerbates under systemic therapy or when a higher dose of systemic therapy causes side effects.

In light of this, we recently found (unpublished data) that the combined utilization of systemic medication, along with a local application of a compound containing lidocaine 2%, pregabalin 5%, and ibuprofen 5%, with or without the addition of amitriptyline 2% Hcl, demonstrates an additive effect in managing oral neuropathic pain, particularly among women. Despite the use of different compounds, our findings align with a previous study that concluded that combined systemic and topical medication therapy is more efficacious for orofacial pain compared to each as a standalone [37].

#### 3.1.1. The Transmucosal Drug-Delivery Systems

The oral mucosa is different from bare skin. There is no stratum corneum; therefore, the drug formulation in relation to adhesive and absorption properties should be different, and other factors should be considered as well, such as taste and the possibility of swallowing. In addition, the salivary washout effect, mucus presence, and constant mouth movements involved in swallowing, speaking, and chewing represent other challenges in transmucosal drug delivery [43]. Muco-adhesiveness may play a major role to be considered in the development of transmucosal drug-delivery systems, increasing drug contact and bioavailability at the permeation site [44].

Given the unique conditions in the oral area, particularly the washing off by salivary flow, the most effective method to apply a topical paste, gel, or ointment is via a custom-made splint, also known as a neurosensory stent or drug delivery device (Figure 1). The topical mixture can be accommodated within this silicone stent, which is designed to confine the mixture dispersion to the afflicted area. It is believed that a combination of several therapeutic substances optimizes the effect [38], even though it has never been rigorously tested. Interestingly, the stent itself may induce significant relief through neurosensory mechanisms, although this claim is primarily based on case series and lacks extensive evidence [45,46]. It is speculated that oral appliances might either diminish sensory load by acting as a physical barrier or, alternatively, serve as a sensory input to inhibit pain pathways, possibly involving placebo mechanisms [45].

However, in addition to the sensory input inhibition characteristics of the stent, we believe that the application of local substances beneath it, primarily different combinations of routinely administered drugs used for neuropathic pain formulated into a topical gel by a pharmacist, is advantageous. Compounding effective preparations for the oral cavity requires certain considerations. First, ensure all active pharmaceutical ingredients (APIs) are fully dissolved by incorporating suitable solvents and excipients. The vehicle should be muco-adhesive with optimal viscosity for prolonged contact; the choice between lipophilic or hydrophilic depends on the API’s nature. In drug compounding, commercially available tablets are often employed in conjunction with various solvents. Prominent among these solvents are glycerin and Tween 80, recognized as polysorbate 80. This is a ubiquitous non-ionic surfactant and emulsifier with applications spanning pharmaceuticals, foodstuffs, and cosmetics. It is characterized by its viscous, water-soluble nature. Additionally, the drug formulation employs a gel called poloxamer 407, a hydrophilic non-ionic surfactant. Most of the common uses of poloxamer 407 are related to its surfactant properties [47]. For example, it is widely used in cosmetics for dissolving oily ingredients in water and also in oral rinses. The gel prepared possesses thermo-reversible properties, meaning its consistency changes based on the temperature. At body temperature, around 37 °C (98.6 °F), it is more viscous, allowing it to adhere better to oral tissues. However, at cooler temperatures, its consistency is more liquid-like. This unique property ensures optimal adhesion to tissues within the oral cavity. The gels in the medication have a high water content, facilitating the optimal dissolution of pregabalin (one of the major drugs used), for example, which is primarily soluble in water. To enhance the palatability of the drug, saccharin, a sweetener, and orange oil, which adds flavor and aroma, are incorporated into the formulation. This not only makes the drug more acceptable to patients but also may improve adherence to the therapy.

Other sweeteners like xylitol and stevia, sucralose, and flavoring oils like peppermint and lemon oil may be considered. All these ingredients are processed together in an electronic mortar and pestle—a modern tool for compounding pharmacies that enables mixing preparations directly in the dispensing container. It is innovatively designed to enhance pharmaceutical compounding and produce consistent preparations [48] (Figure 2).

#### 3.1.2. Proposed Intra-Oral Topical Compounds for Neuropathic Pain

The data regarding drug combinations used inside the mouth for neuropathic pain are very limited, lacking head-to-head studies and standardization. They are mostly found in case presentation reports and series and expert opinion reviews. These data have been summarized in Table 2.

Topical preparations of commonly known systemic drugs for neuropathic pain that have been suggested for intraoral use are ketamine 4%, carbamazepine 4%, lidocaine 1%, ketoprofen 4%, gabapentin 4%, and especially pregabalin 5–10% within a lipoderm inert vehicle. The method of use is usually beneath a neurosensory stent 3 to 4 times per day [23].

An expert opinion review suggests that the first-line topical medication options for trigeminal neuropathies include a topical anesthetic and/or capsaicin. The anesthetics include mostly benzocaine and lidocaine; the application may be repeated 4 to 6 times per day [49]. Vickers et al. treated patients diagnosed with atypical odontalgia with an application of 0.025% capsaicin for 3 min twice a day, after local anesthetic application to the painful site, with pain relief for 19 out of 50 patients after 1 month [50]. However, treatment adherence to capsaicin is low due to its side effects, particularly the burning pain associated with its application [52]. A retrospective study on 39 patients with varied orofacial neuropathic conditions, such as post-traumatic neuropathies and trigeminal neuralgia, examined the efficacy of topical medications as standalone or adjunctive treatment with systemic medications. The combination used included 4% carbamazepine, 1% lidocaine, 4% ketoprofen, 4% ketamine, and 4% gabapentin, which significantly reduced pain levels, particularly in the group receiving combined therapy (systemic and topical) [37]. A study conducted on rats with local neuropathic pain applied a topical regimen of pregabalin 10% and diclofenac 5% observed a significant reduction in pain intensity along with a notable decrease in drug plasma concentration compared to systemic administration of these drugs [51]. A case report by Haribabu et al. implemented a slightly different drug combination, consisting of 4% carbamazepine, 1% lidocaine, and 4% gabapentin, reported similar findings of substantial pain reduction following several weeks of topical treatment [34]. As mentioned earlier, concentrations and combinations varied among experts and institutions.

## 4. Conclusions

The utilization of topical treatments, especially under custom-made silicone splints, presents a promising approach for the management of chronic neuropathic intra-oral pain. These interventions offer targeted relief and are generally associated with minimal adverse effects, making them a preferable option in certain clinical scenarios. The effectiveness of these methods appears to be enhanced when a range of different analgesic agents are incorporated, suggesting a benefit in utilizing a diverse array of pain-relieving strategies. In particular, topical applications play a vital role as an “escape drug” or an additional treatment method, especially when patients experience exacerbations of pain under systemic therapy or when higher doses of systemic treatments lead to undesirable side effects.

Despite these promising aspects, the current landscape of research in this area is fraught with limitations. The bulk of the evidence stems from expert opinions rather than large-scale empirical studies. Many of the studies conducted to date suffer from limited sample sizes, which can impact the generalizability of their findings. Additionally, there is a notable inconsistency in the methodologies employed across different studies, which further complicates the ability to draw firm conclusions. Another significant issue is the lack of extended follow-up data, which is crucial for understanding the long-term efficacy and safety of these interventions. Given these gaps in the current body of research, there is a pressing need for more comprehensive and well-structured studies. For example, sensory stents containing only the drug vehicle compared against those containing active drugs, may shed light on the mechanical effects of pain reduction of these devices.

Future research endeavors should aim to increase sample sizes to improve the reliability and applicability of the results. There is also a need for standardization in study methodologies to ensure consistency and comparability across different research efforts. Long-term follow-up is essential to assess not only the immediate effectiveness of these treatments but also their long-term impact, potential side effects, and overall patient outcomes.

Take home message:Topical treatments for oral neuropathic pain show potential but require stronger evidence for efficacy;Synergistic benefits might exist in combined topical formulations, warranting further investigation into their underlying mechanisms;While stents improve drug delivery to affected areas, their independent advantages remain unclear.

## Figures and Tables

**Figure 1 pharmaceuticals-17-00264-f001:**
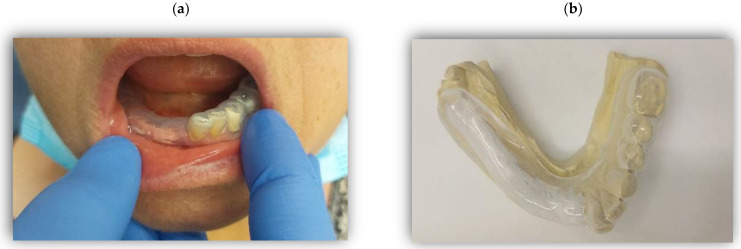
(**a**,**b**)—Custom-made splint or neurosensory stent for delivering topical therapeutics to alleviate neuropathic pain, targeting the lower mandible gums. (**a**)—in mouth. (**b**)—on model.

**Figure 2 pharmaceuticals-17-00264-f002:**
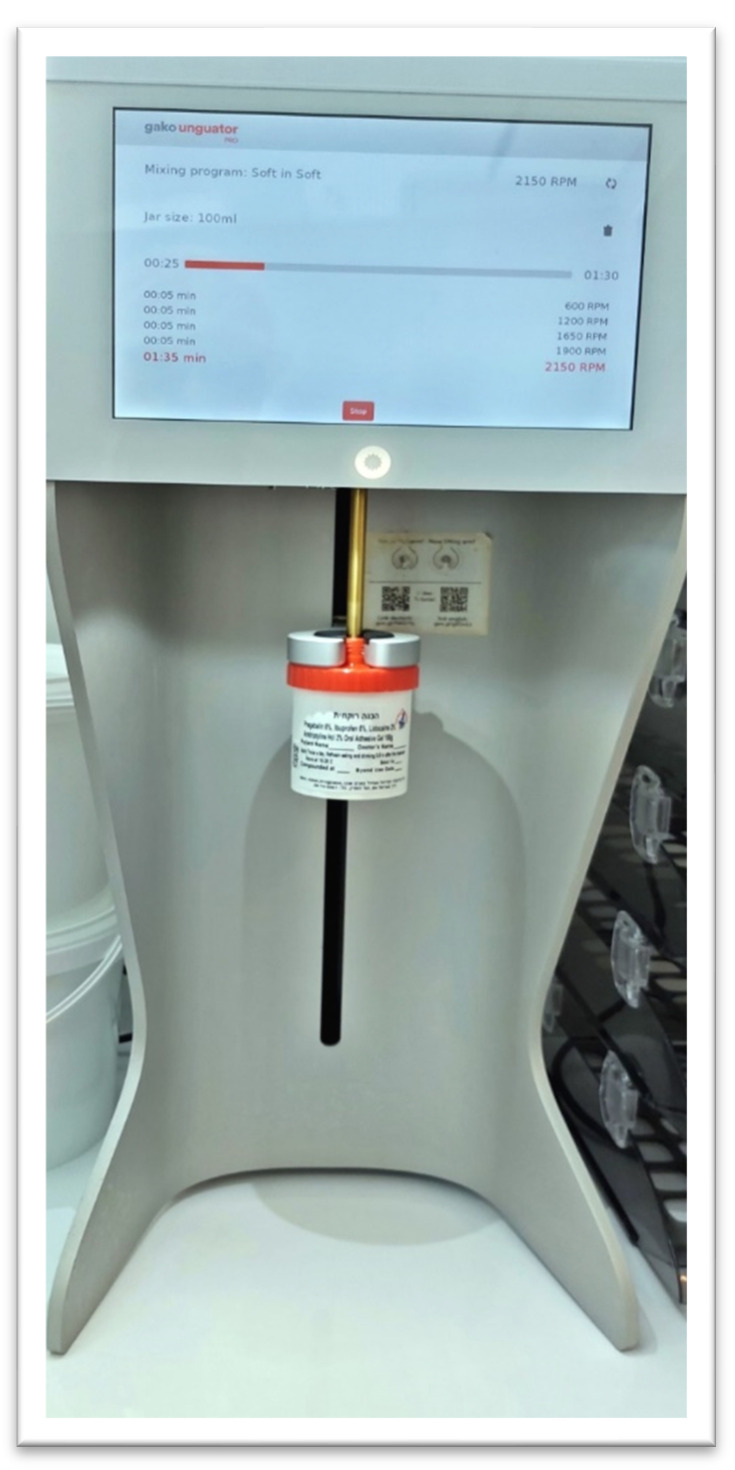
Electronic mortar and pestle: A modern tool essential for compounding pharmacies, this enables mixing preparations directly in the dispensing container. It is innovatively designed to enhance pharmaceutical compounding, produce refined preparations, and reduce nonproductive time (Courtesy pharmacist Eyal Zur).

**Table 1 pharmaceuticals-17-00264-t001:** Common Drugs for Neuropathic Pain—Targets, Actions, and Systemic Use Side Effects [24,25].

Drug	Direct/Indirect Mechanism of Action	Cellular Targets	Medical Considerations (with Systemic Use)	Common Side Effects (with Systemic Use)
*Systemic/topical route* **Carbamazepine**	GABA system modulation, Sodium-channel blockade, NMDA-receptor blockade	Neurons Keratinocytes	Bone marrow problems, cardiac problems, blood disorders, glaucoma	Hypertension, hypotension, lightheadedness, rash, pruritus, erythematous condition, nausea, vomiting, confusion, dizziness, nystagmus, somnolence, blurred vision, diplopia
**Gabapentinoids:** **Gabapentin** **Pregabalin**	Calcium-channel blockade, NMDA blockade, Potassium-channel activation, Anti-inflammatory effect	Neurons Keratinocytes	Renal problems	Edema, myalgia, ataxia, dizziness, somnolence, tremor, mood swings, hostility, fatigue, weight gain, constipation, xerostomia, blurred vision, diplopia
**Amitriptyline**	Sodium-channel blockade, Calcium-channel blockade, NMDA receptor antagonism, Potassium-channel activation, Adrenergic receptor down-regulation, TRPA1 desensitization, GABA receptor modulation, 5-HT receptor blockade, Histamine receptor blockade, Anticholinergic, Reduction in nitric oxide, prostaglandin E2, Noradrenaline, 5-HT, dopamine and adenosine reuptake inhibition, Opioid system modulation	Neurons Keratinocytes	Cardiac problems, diabetes, epilepsy, glaucoma, urinary retention, thyroid problems, hepatic problems, psychiatric problems	Weight gain, somnolence, blurred vision, fatigue, urinary retention, headache, asthenia, dizziness, constipation, xerostomia
**Anti-inflammatory:** **Diclofenac** **Ketoprofen**	Cox-2 inhibition, NMDA receptor antagonism, TRPV1, TRPA1, TRPM3 ligand, Sodium-channel blockade, Calcium-channel inhibition, Potassium channels modulation, Adrenergic receptor interaction, Opioid receptor modulation	Immune cells Neurons Keratinocytes Schwann cells	Cardiovascular problems, gastrointestinal disease, asthma	Edema, gastrointestinal complaints, rashes or pruritus, tinnitus, dizziness, somnolence, headache, increased liver function tests
*Topical route* **Lidocaine**	Sodium-channel blockade, Cholinergic-receptor blockade, TRPA1desensitization, NMDA receptor antagonism, Acid-sensing ion channel blockade, Potassium-channel blockade, Calcium-channel blockade, P2X7 inhibition, Nerve growth factor modulation, Glycinergic pathways modulation, Anti-inflammatory effect	Neurons Keratinocytes Immune cells Schwann cells	None (with topical use)	Local irritation, erythema
**Capsaicin**	TRPV1 activation	Neurons Keratinocytes	None	Local burning sensation, erythema

**Table 2 pharmaceuticals-17-00264-t002:** Summary of Current Literature on Intra-Oral Topical Treatment Options.

Reference	Summary	Drugs Mentioned	Method of Use
[34]. Haribabu et al., 2013	Case report on topical medications for managing neuropathic orofacial pain, including various drug combinations.	4% carbamazepine, 1% lidocaine, 4% gabapentin	Several weeks of topical treatment
[35]. Nasri-Heir et al., 2013	Overview of topical medications as a treatment for neuropathic orofacial pain, highlighting their application and benefits.	N/A	N/A
[1]. Padilla et al., 2000	Review of topical medications for orofacial neuropathic pain, discussing their applications and potential effectiveness.	N/A	N/A
[38]. Dworkin et al., 2007	Evidence-based recommendations for pharmacologic management of neuropathic pain, including topical options.	N/A	N/A
[10]. Baad-Hansen and Benoliel, 2017	Discussion of neuropathic orofacial pain, highlighting facts and fiction in its treatment approaches.	N/A	N/A
[23]. Heir et al., 2022	A narrative review discussing the use of compounded topical medications for orofacial pain treatment.	ketamine 4%, carbamazepine 4%, lidocaine 1%, ketoprofen 4%, gabapentin 4%, and especially pregabalin 5–10% within a lipoderm inert vehicle	Application beneath neurosensory stent 3 to 4 times per day
		capsaicin 0.025–0.075% paste or 8% patch	Up to 4 times a day for 8 weeks 1–4 patches for 30–60 min, once in 3 months
[41]. Casale et al., 2014	Case report on the use of 5% lidocaine-medicated plaster for localized neuropathic orofacial pain.	5% lidocaine-medicated plaster	Application to affected area
[42]. Fusco and Alessandri, 1992	Discussion of the analgesic effect of capsaicin in trigeminal neuralgia.	capsaicin	Application to affected area
[45]. Bavarian et al., 2022	Retrospective case series on the use of oral appliances in the management of neuropathic orofacial pain.	N/A	N/A
[46]. Axell, T., 2008	Exploration of the treatment of painful symptoms in the oral mucosa using lingual acrylic splints.	N/A	N/A
[49]. Patel S et al., 2018	Discussion of topical medications for common orofacial pain conditions, emphasizing practical applications.	benzocaine, lidocaine	N/A
[50]. Vickers et al., 1998	Analysis of patients with atypical odontalgia and pharmacological procedures for diagnosis and treatment.	0.025% capsaicin for 3 min twice a day after local anesthetic application	Local application for 3 min twice a day after local anesthetic application
[37]. Heir et al., 2008	Retrospective study on the use of topical medications in orofacial neuropathic pain treatment.	4% carbamazepine, 1% lidocaine, 4% ketoprofen, 4% ketamine, and 4% gabapentin	Topical application
[51]. Plaza-Villegas et al., 2012	The only study on an animal model—rats with local neuropathic pain, applying a topical regimen of pregabalin and diclofenac—showing a reduction in pain intensity.	Topical regimen of pregabalin 10% and diclofenac 5%.	Topical application

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
