# Peer review of "Pharmacological Topical Therapy for Intra-Oral Post Traumatic Trigeminal Neuropathic Pain: A Comprehensive Review"

_pharmaceuticals, 2024, doi:10.3390/ph17020264_

Round 1

Reviewer 1 Report

Comments and Suggestions for Authors

Thank you for the opportunity to review this work.

However, the manuscript still has the following problems worthy of attention, through the improvement of these problems can better improve the quality of the manuscript.

Abstract

Introduction: The background in the abstract could be shortened to 1-2 sentences that highlight the key context. The objectives need to be stated explicitly at the end of the intro.

Methods: Add 1-2 sentences summarizing the methods - literature search strategy, databases searched, inclusion/exclusion criteria etc.

Results: The results section in the abstract should concisely summarize the key findings from the literature review, without interpretation. Quantify results where possible e.g. X studies reported Y result, Z studies showed opposite results.

Conclusion: Interpretation of results and implications should be reserved for the conclusion section. The take-home message and significance of the review should be highlighted. Conclusion can state 2-3 major implications of the findings.

Introduction

Condense the background to 1-2 paragraphs focused on key context of neuropathic pain and topical treatments. Remove details not directly relevant.

Clearly state the rationale and knowledge gaps that led to this review - why is it important/needed?

Explicitly state the objectives and scope early in the introduction. What questions is this review trying to answer?

Use more topic sentences to transition between background and objectives.

Avoid repetitiveness. For example, the pathophysiology of neuropathic pain is described in both the background and later in the post-traumatic neuropathic pain section.

Focus on concise and clear writing, remove convoluted or redundant phrasing.

Use more references to back up statements in the background/context.

End the introduction with a paragraph clearly stating the rationale, objectives and scope to provide direction.

Methods

The methods section is missing in this paper. Add a methods section that describes how the literature review was conducted, including details on the literature search strategy, databases searched, inclusion/exclusion criteria, screening process etc. This will make the methodology more robust and transparent.

Results/Discussion

The results and discussion are combined into one section in this paper. Consider separating into distinct results and discussion sections for better organization.

The results should focus on summarizing the key findings from the literature synthetically and objectively, without interpretation. Reserve interpretation for the discussion section.

The discussion could be strengthened by comparing and contrasting findings between studies, analyzing consistencies/inconsistencies in results, providing explanations for contradictions etc. This will provide a more critical analysis.

Discussion can also highlight study limitations identified in the reviewed literature and suggest future research directions to address the gaps.

2. Orofacial neuropathic pain types

- Consider reorganizing this section for better flow. For example:

1) Introduction to orofacial neuropathic pain types

2) Description of relevant conditions (TN, PHN, BMS etc.)

3) Pathophysiology

4) Focus on post-traumatic neuropathic pain

Special

- Avoid repetition between sections - for example pathophysiology is covered twice.

- Remove details not directly relevant to the review focus.

- Add more topic sentences to transition between concepts.

- Clarify which orofacial pain types are within scope and which are not.

- Eliminate convoluted and redundant phrasing for clarity.

- Use more references to substantiate statements.

- Reduce wordiness and condense concepts that are well established.

- Consider using tables/figures for treatment approaches or medication mechanisms if helpful for reader comprehension.

- Make sure to connect back to main review objectives at the end.

3. Neuropathic pain topical management 

- Clearly state the scope of this section - which topical treatments are being reviewed and why.

- Consider reorganizing subsections for better flow:

1) Introduction to topical treatments

2) Mechanisms and advantages

3) Evidence for topical treatments in general neuropathic pain

4) Oral applications

5) Challenges with oral topical delivery

6) Proposed formulations for oral neuropathic pain

Special

- Avoid repetitive information between sections.

- Condense general topical treatment background to 1-2 paragraphs - focus on key mechanisms and evidence.

- Provide more details on proposed oral topical formulations - active ingredients, synergistic mechanisms, evidence of efficacy if available.

- Use topic sentences and transitions to improve flow between concepts.

- Add references to substantiate statements.

- Clarify which oral topical formulations are within scope. Avoid digressing into general medications.

- Consider adding a table summarizing current evidence on oral topical treatments.

- Evaluate if subsections could be condensed or removed if not directly relevant.

- Make sure to connect back to main review objectives.

Conclusion

The conclusion summarizes the existing evidence but does not highlight the implications, overall significance or 'take-home message' of the review. Adding this will make the conclusion more impactful.

Suggest including 3-4 bullet points that succinctly state the key conclusions and implications of this review.

Clearly state the implications arising from this review. For example:

1) Topical treatments seem promising for oral neuropathic pain but better quality evidence is needed.

2) Combination topical formulations may have synergistic effects but more research on mechanisms is needed.

3) Stents enhance drug delivery but their standalone benefits are uncertain.

Special

- Use concise, clear language and avoid redundancies.

- Make sure the conclusion provides a logical sense of closure for the reader.

General

The paper would benefit from proofreading to correct minor grammatical/language errors.

Add a limitation section acknowledging the limitations of this literature review.

Comments on the Quality of English Language

The paper would benefit from proofreading to correct minor grammatical/language errors.

Reviewer 2 Report

Comments and Suggestions for Authors

In the current  review are examined the current studies concerning the efficacy of topical  treatments for neuropathic pain within the oral cavity, with a particular focus on post traumatic trigeminal neuropathic pain. The authors underline that despite preliminary evidence suggesting the effectiveness of topical treatments for oral neuropathic pain, further studies are needed to confirm these treatments efficacy, standardize methodologies, and establish long-term outcomes. The aim of the study  is to offer a clearer direction for managing chronic neuropathic pain in  the oral cavity.

Some suggestions:

1. Table 1.:

-Add please the recommendations concerning the mode of administration for each drug (route of administration, amount administered, no. and duration of administrations).

-Please also clarify the aspects related to the topical application and the head of the Table 1: Medical considerations (with systemic use) and Common side effects (with systemic use).

-The reference are missing for Table 1. Please add.

2. Neuropathic pain topical management, lines 163-166:  It would be interesting to present such a bioavailability study, but not on healthy patients.

3. Lines 205-209:

-line 206: What systemic medication was used? Please add. -line 207-209: Do you can explain why you consider that the additive effect is better among women? Please add.   4. Lines 258-59: To enhance the palatability of the drug, can other sweeteners and oils be used besides saccharin and orange oil? Plese add.  

5. Table 2: For each study you must add if the study was performed on humans or on rats/mice. Also, in each case, you must specify which pharmaceutical form was applied (gel, cream?).

Round 2

Reviewer 1 Report

Comments and Suggestions for Authors

None